# The Coupling Effect of Organosilicon Hydrophobic Agent and Cement on the Water Resistance of Phosphogypsum

**DOI:** 10.3390/ma15030845

**Published:** 2022-01-22

**Authors:** Pengfei Ma, Chong Wang, Yuxin Gao, Xiaowei Gu, Baojun Cheng, Zheng Fang, Guangqi Xiong, Jing Wu

**Affiliations:** 1College of Materials Science and Engineering, Chongqing University, Chongqing 400045, China; mapengfei_051@163.com (P.M.); 20133270@cqu.edu.cn (Y.G.); fang_zheng@cqu.edu.cn (Z.F.); xgqcqu@163.com (G.X.); 2Building Materials Science Academy of China West Construction Group Co., Ltd., Chengdu 610213, China; Baojuchen2022@163.com; 3College of Resources and Civil Engineering, Northeastern University, Shenyang 110819, China; guxiaowei@mail.neu.edu.cn; 4College of Materials Science and Engineering, Wuhan Textile University, Wuhan 430200, China; 13452334306@163.com

**Keywords:** phosphogypsum, water-resistance, mechanical properties, softening coefficient, hydrophobic film

## Abstract

The objective of this paper is to investigate the coupling effect of cement and organosilicon hydrophobic agents on the water resistance of phosphogypsum. Different weight ratios of Portland cement were added to adjust the alkalinity of this system and further improve the work efficiency of the organosilicon hydrophobic agents. Some macroscopic performances, such as the water absorption, the compressive strength, the flexural strength, and the softening coefficient, were measured to characterize the water-resistance of phosphogypsum. The microscopic characteristics were analyzed via contact angle tests, scanning electron microscopy (SEM) and X-ray photoelectron spectroscopy (XPS) to understand the mechanism of organosilicon hydrophobicity. The results indicated that both the compressive and flexural strengths of phosphogypsum first increased and then decreased with the increase of organosilicon hydrophobic agents. Meanwhile, the surface contact angle continued to increase and the softening coefficient exhibited an obvious increase. When the hydrophobic agent was combined with Portland cement, the softening coefficient of phosphogypsum further increased from 0.80 to 0.99, while the water absorption rate was significantly reduced from 16.0% to 0.8%. Microscopic tests proved that the hydrophobic organic molecules can be polymerized under the high alkalinity, and promote the formation of a hydrophobic film, thus significantly improving the water-resistance of phosphogypsum.

## 1. Introduction

Phosphogypsum is a massive amount of solid waste in southwest China. The storage of phosphogypsum not only encroaches on the land, but also causes significant environmental pollution [1]. Although some attempts have been made to use phosphogypsum as a wallboard and building blocks in the field of construction [2,3,4], the poor water-resistance of phosphogypsum severely restricts its widespread application [5]. Many scholars have carried out researches to solve this problem. In recent years, hydrophobic materials such as silane, siloxae, fluoropolymers and silicon resin have obtained great attention. Fluoropolymers were bonded to the substrate by an interface layer, while alcohols and organosilicon were bonded to the substrate surface, mainly by hydrolysis and a polycondensation reaction. Zhu et al. [6] added polyvinyl alcohol to make the phosphogypsum denser and improve the water resistance. Sun et al. [7] applied an impermeable paraffin emulsion to the surface of the gypsum to prevent water migration. An et al. [8] added a polyvinyl alcohol-polypropylene emulsion to form a hydrophobic layer over the pores of the gypsum to make it waterproof. Hydrophobic membranes have a small influence on the strength of the gypsum while decreasing gypsum’s permeability significantly [9,10]. In order to broaden the prospect of solid waste utilization of phosphogypsum, it is necessary to overcome the characteristics of poor water durability. Thus, it is a feasible idea to improve the water-resistance of phosphogypsum. The hydrophobic agent can be used to form hydrophobic membranes on the plaster surface. The modified plasters have excellent water-resistance in the case of a low weight ratio of hydrophobic agents. On the other hand, the alkalinity of cement can change the hydrolysis and condensation chemistry of organosilicon compounds [9,10]. Therefore, it may be a good idea to prepare waterproof phosphogypsum by adding a small amount of cement. However, few studies systematically analyzed the coupling effect of cement and organosilicon hydrophobic agents on the water-resistance of phosphogypsum.

This paper is organized into three sections. Section 1 introduces the background, purpose and novelty of this paper. Section 2 describes the details of raw materials, the mixing proportion and test methods, the modification of phosphogypsum with different organosilicon hydrophobic agent content, and the coupling of 0.3% organosilicon hydrophobic agent and cement content was systematically studied. The macroscopic performance of phosphogypsum, including the fluidity, the setting time, the water-resistance and the mechanical properties, were systematically investigated. Moreover, the surface contact angle experiment was used to characterize the degree of hydrophobicity. In addition, the X-ray photoelectron spectroscopy (XPS) test was used to explore the action mechanism of the silicone hydrophobic agent under different alkalinity environments. The microscopic morphology of phosphogypsum was conducted by SEM qualitatively. Section 3 presents the experimental results and discussion. It would be substantiated that the coupling of an organosilicon hydrophobic agent and cement content is a more effective way to modify the water-resistance of phosphogypsum. The enhancement of phosphogypsum’s water-resistance can help accelerate the process of phosphogypsum solid waste utilization.

## 2. Methodology

### 2.1. Raw Materials

Phosphogypsum was used as an industrial by-product of the phosphorous gypsum powder, with a density of 2.52 kg/m^3^. The ordinary Portland cement 42.5R (P.O. 42.5R), with a specific surface area of 360 m^2^/kg, was used in this study. The organosilicon hydrophobic agent produced by China State Construction Engineering Co., Ltd. (Chengdu) was used to improve the water-resistance performance. The basic molecular structure is shown in Figure 1. Organic groups were arranged on the surface of hydrophobic film. It reacted with by droxyl groups in silicate substrate to form siloxane chains, thus bridging the surface of cement matrix and preventing the invasion of external water. The polycarboxylic acid type superplasticizer, supplied by the Western China Construction Materials Science Research Institute, was used as a water reducing agent, with a solid content of 25%.

The chemical components of phosphogypsum and cement were tested by X-ray fluorescence (XRF) and are shown in Table 1.

### 2.2. Mixing Proportion

The mix proportions of phosphogypsum are detailed in Table 2. The hydrophobic agent content was 0 wt.%, 0.1 wt.%, 0.3 wt.%, and 0.5 wt.% of phosphogypsum. The alkalinity of the gypsum hydrophobic agent system was adjusted by the weight ratios of the cement to improve the hydrophobic effects. In the coupling cases of hydrophobic agents and cement, the fixed hydrophobic agent content was 0.3 wt.% and the cement was mixed with 0.2 wt. %, 0.4 wt.%, and 0.6 wt.% instead of phosphogypsum.

### 2.3. Test Methods

#### 2.3.1. Water-Resistance

After curing for 7 d, six 40 mm × 40 mm × 160 mm prism blocks were used to test the water absorption, the flexural strength and compressive strength. Three 40 mm × 40 mm × 160 mm prism blocks were dried to a constant weight in a 45 °C vacuum-desiccant box and then cooled to room temperature after being removed. The softening coefficient, the water absorption and the compressive and flexural strength were tested systematically based on the GB/T 23451-2009 Gypsum Block [11]. Softening coefficient (I) is the ratio of compressive strength in a water-saturated state (*σ*_1_) and compressive strength in a dry state (*σ*_0_), as expressed in Equation (1).
(1)$$I=\frac{\sigma_{1}}{\sigma_{0}}$$

The water absorption test was performed according to the standard JCJ 70-2009 “Basic Performance Test Method for Building Mortar” [12]. The compressive and flexural strength tests were carried out in accordance with the standard GB/T17671–1999 “Cement Mortar Strength Test Method (ISO method)” [13].

The contact angle was tested by a CA-100D automatic contact angle measuring instrument, which was produced by Shanghai Shanghai Jiazun Instrument Equipment Co., Ltd. in China, according to the standard GB/T 30447-2013 “Measurement method for contact angle of nano-film surface” [14]. The size of the tested specimen was 40 mm × 40 mm × 160 mm. After drying at 45 °C to a constant weight, the specimens were removed and cooled to a constant temperature at 22 °C. The fractured gypsum specimens were loaded onto a flexural strength test instrument, and a high-resolution camera observed the changes in at a water droplets contact time of 60 s.

#### 2.3.2. Microstructure

The microstructural samples were immersed into absolute ethyl alcohol solution to terminate hydration, and dried to a constant weight in the vacuum drying oven at 45 °C. The SEM samples were examined using a TESCAN VEGA2 scanning electron microscope, which was produced by Tescan (China agent) Co., Ltd. in the Czech republic, with an acceleration voltage of 20.0 kV. The XPS samples were tested with a Thermo Fly Escalab 250 Xi with a full-spectrum throughput of 100 eV, with a step size of 1.0 eV, and a narrow spectrum of 30 eV, with a step size of 0.05 eV. The vacuum in the analysis area was <2.7 × 10^−10^ mbar.

## 3. Results and Discussion

### 3.1. Influence of the Hydrophobic Agent Content on the Properties of Phosphogypsum

#### 3.1.1. Fluidity and Setting Time

Table 3 shows the fluidity and setting time of the phosphogypsum with different dosages of a hydrophobic agent.

As shown in Table 3, with an increasing amount of silicone hydrophobic agent, the fluidity of gypsum was slightly improved, with little difference in the working performance. The initial and final setting times were slightly improved because the silicone hydrophobic agent did not affect the crystallization and growth of the gypsum crystals by adsorbing on the surface of the crystals.

#### 3.1.2. Hydrophobic Agent Water-Resistance and Mechanical Properties

The effects of the silicone hydrophobic agent at different dosages on the softening coefficient, water absorption rate, compressive strength and flexural strength of phosphogypsum at 7 d were studied. The results are shown in Figure 1.

Figure 2a showed that when the hydrophobic agent content was 0–0.5 wt.%, the softening coefficient of the phosphogypsum increased significantly from 0.50 to 0.99, as the content of the hydrophobic agent increased. Water absorption decreased from 18.0% to 0.9%. This was because the hydrophobic agent was adsorbed onto the surface of the gypsum crystals, and hydrophobic groups faced outward, forming a hydrophobic film and improving the water-resistance of the phosphogypsum effectively [15].

Figure 2b showed that, compared with that of the reference group, the compressive strength of the samples decreased as the hydrophobic agent decreased. This was because the hydrophobic agent was adsorbed onto the surface of the gypsum crystals, hindering the hydration of gypsum and leading to a decrease in the compressive strength. When the amount of hydrophobic agent was 0–0.5 wt.%, the flexural strength of gypsum first decreased and then increased with the increment of the hydrophobic agent. This may be because when the amount of added hydrophobic agent was low, the hydrophobic film was not completely formed on the gypsum crystal surface. The small amount of film adsorption on the surface of the gypsum crystal, limited the hydration reaction and the formation of gypsum, leading to the reduction in flexural strength. With an increasing amount of hydrophobic agent, a complete water-repellent film was formed on the surface of the gypsum paste. This film had a favorable toughening effect and increased the flexural strength greatly.

However, within the range of 0.1–0.5 wt.% of an added hydrophobic agent, the compressive strength gradually increased with increasing hydrophobic agent content. This is because, with the increasing amount of added hydrophobic agent, a complete hydrophobic film was formed on the surface of gypsum crystals, which gradually reduced the influence of gypsum crystal formation [16]. In addition, some hydrophobic agents filled the voids in the gypsum matrix and compacted the matrix, enhancing the strength of the gypsum block.

### 3.2. The Influence of Cement Content on the Physical and Mechanical Properties

#### 3.2.1. Fluidity and Setting Time

Table 4 showed the extremely fast fluidity and setting times of phosphogypsum with different cement contents.

As shown in Table 4, by increasing the cement content, the fluidity of gypsum increased because the water demand for cement was less than that of gypsum. The setting time of the cement-gypsum system increased with an increasing amount of added cement because the setting time of cement was longer than that of gypsum (4 min for initial setting and 8 min for final setting).

#### 3.2.2. Water Resistance and Mechanical Properties of Hydrophobic Agents

The alkalinity of the gypsum hydrophobic agent system was adjusted with cement, and the fixed hydrophobic agent content was 0.3 wt.%. The influence of cement content on the softening coefficient, the water absorption rate, the compressive strength and the flexural strength of gypsum specimens with a hydrophobic agent were studied. The results are shown in Figure 2.

Figure 3a showed that when the weight ratio of cement was in the range of 0–0.6 wt.%, the gypsum softening coefficient increased gradually as the weight ratio of cement increased, and the water absorption rate was significantly reduced. This was due to the increased weight ratio of cement and the subsequent alkalinity in the cement-phosphogypsum system, promoting the hydrolysis of the hydrophobic agent, as well as polymerization, and then causing the hydrophobic agent to form a hydrophobic film on the gypsum crystal surface. This film prevented moisture from entering, improving the phosphogypsum water resistance, which was in accordance with Su’s research [17].

When the cement content was 0.6 wt.%, gypsum had the best water-resistance performance. In this situation, the softening coefficient was 0.99, the water absorption rate was 0.8%, and the pH of the slurry was 7–8. Figure 3b showed that the increase in cement admixture lowered the mechanical properties of the gypsum block; namely, the compressive strength and flexural strength further decreased because the cement promoted the hydrolysis of the organosilicon hydrophobic agent and the polymerization reaction of the gypsum crystals. More organic silicon hydrophobic agent was added to the surface of the gypsum slurry, inhibiting the gypsum hydration reaction and interfering with the gypsum crystal structure, leading to a decrease in the strength of the gypsum block [18].

### 3.3. The Influence of Cement and Hydrophobic Agents on the Surface Contact Angle of Phosphogypsum

The influence of different dosages of added silicone hydrophobic agent and different weight ratios of cement on the surface contact angle of the phosphogypsum block was studied, and the results are shown in Figure 4. As Figure 4a depicts, the contact angle of the phosphogypsum reference group was the smallest at only 11.57°, which was due to the high hydrophilicity, high water absorption and poor water-resistance of the surface of the gypsum block. As Figure 4b–d shows, the amount of silicone hydrophobic agent ranged from 0.1 wt.% to 0.5 wt.%, and the gypsum contact angle gradually increased, indicating the favorable hydrophobic property of the gypsum block and that the surface gradually changed from hydrophilic to hydrophobic. As Figure 4d shows, when the amount of silicone hydrophobic agent was 0.5 wt.%, the surface contact angle of the phosphogypsum block reached 112.56°, which suggests that the gypsum block has good waterproofing properties.

As Figure 4c shows, when the amount of silicone hydrophobic agent in the phosphogypsum block was 0.3 wt.%, the surface contact angle was 47.56°. As Figure 4e–g shows, the addition of cement can increase the surface contact angle of the phosphogypsum block. When the cement content was 0.2–0.6 wt.%, the surface contact angle of the phosphogypsum block was greater than 120°. By increasing the cement content, the surface contact angle increased further. When the cement content was 0.6 wt.%, the phosphogypsum block surface contact angle reached 147.23°, indicating excellent water resistance. The reason was that when cement was added to the phosphogypsum system, the alkalinity of the cement-phosphogypsum system was increased, and the hydrophobic film formed on the surface of gypsum by the hydrophobic agent was improved because it had an increased density and tended to be complete. Therefore, the phosphogypsum block became more hydrophobic, and the surface contact angle of the phosphogypsum block increased.

### 3.4. Influence Mechanism of Hydrophobic Agent and Cement

#### 3.4.1. Hydration Products of Phosphogypsum

To further explore the influence mechanism of the dosages of the silicone hydrophobic agent and the weight ratios of cement on the water resistance performance of the phosphogypsum system, XPS was carried out by selecting four mixture ratios: the original solution of the hydrophobic agent, the reference group of phosphogypsum, phosphogypsum +0.3% hydrophobic agent, and phosphogypsum +0.3% hydrophobic agent +0.6% cement. Figure 5 shows the total XPS spectrum of the elements in the gypsum samples, and the content of each element is shown in Table 5.

Both the XPS spectra in Figure 5 and the element content in Table 5 showed that the main elements of the silicone hydrophobic agent are C, Si and O, and that Si and O account for 33.91% and 56.52% of the substance, respectively, indicating that the hydrophobic agent contained a large number of Si-O bonds and OH^-^. The main elements in the phosphogypsum, phosphogypsum +0.3 wt.% hydrophobic agent, and phosphogypsum +0.3 wt.% hydrophobic agent +0.6 wt.% cement were O, C, S, Ca and Si (in order of descending quantity). The C content in the gypsum reference group was 21.69% because the water-reducing agent and retarder used contained C. It indicated that the hydrophobic agent was adsorbed onto the surface of the gypsum crystal, increasing the content of C. In addition, the reference group of the phosphogypsum contained 2.57% Si, and the composition analysis of the gypsum raw materials also revealed a small amount of SiO_2_, which was caused by impurities. The Si and C contents of the phosphogypsum +0.3 wt.% hydrophobic agent increased, compared with those of the reference group of phosphogypsum, indicating that the silicone hydrophobic agent was adsorbed onto the surface of the gypsum crystal. Compared with that of the reference group of the phosphogypsum +0.3 wt.% hydrophobic agent +0.6 wt.% cement, the content of Si and C increased more significantly, indicating that the appropriate alkalinity promoted the absorption of the hydrophobic agent to the surface of gypsum, which was conducive to the improvement of the water-resistance of phosphogypsum

In the XPS results, the changes in Si 2p and the peak fitting of O 1s in the phosphogypsum, phosphogypsum +0.3 wt.% hydrophobic agent, phosphogypsum +0.3 wt.% hydrophobic agent and phosphogypsum +0.3 wt.% hydrophobic agent +0.6 wt.% cement are shown in Figure 6. Figure 6a shows that the addition of the hydrophobic agent significantly increased the relative content of silicon in gypsum compared with that of the phosphogypsum reference group. This was because the hydrophobic agent used was an organosilicon hydrophobic agent, and silicon was its main element. The addition of an organosilicon hydrophobic agent to the surface of gypsum crystals increased the content of Si in gypsum. The addition of the hydrophobic agent shifted the peak position of the silicon peak in the gypsum to lower binding energy, indicating that a chemical reaction occurred between the silicone hydrophobic agent and the gypsum, changing the binding state of silicon within the XPS spectrum of the phosphogypsum +0.3 wt.% hydrophobic agent +0.6 wt.% cement or the phosphogypsum +0.3 wt.% hydrophobic agent. The relative content of Si 2p was higher, and the peak position shift was more obvious, indicating that the appropriate alkalinity enhanced the chemical reaction between the hydrophobic agent and gypsum [19].

Figure 6b–d shows the peak fitting results of the O 1s XPS spectra of the three samples, the phosphogypsum +0.3 wt.% hydrophobic agent and phosphogypsum +0.3 wt.% hydrophobic agent +0.6 wt.% cement, respectively. Figure 6b showed that the O 1s spectrum of the phosphogypsum reference group consisted of 78.74% OH^−^ and 21.26% H_2_O, indicating that the hydration product of gypsum contained a large number of hydrophilic groups OH^−^ on the surface of the calcium sulfate dihydrate crystals, which was also one of the main reasons for the high water absorption rate of gypsum. After adding the 0.3 wt.% hydrophobic agents to phosphogypsum, the content of OH^−^ decreased by 9.3%, and the lattice oxygen O_2_^−^ increased by 5.56%, indicating that the hydrophobic agent participated in the OH^−^ polymerization reaction on the surface of gypsum crystals and adhered to the surface of gypsum crystals, forming a continuous water-repellent film and improving the water resistance of gypsum. The phosphogypsum +0.3 wt.% hydrophobic agent +0.6 wt.% cement was compared with the phosphogypsum benchmark group. The 21.60% increase in OH^−^ content and the 12.57% increase in lattice oxygen O_2_^−^ showed that the surface of the gypsum crystals with suitable basicity due to the higher levels of OH^−^, reacts more effectively with the hydrophobic agent. With increased basicity, the hydrophobic agent on the surface of the gypsum crystals increased, and the formation of the hydrophobic membrane was more complete; therefore, the gypsum contact angle was larger, and the material was more resistant to water. Thus, the organosilicon hydrophobic agent adhered to the surface of the gypsum crystal through the polymerization of OH^−^ on the surface of the gypsum. When the cement was added together with a hydrophobic agent, the appropriate alkalinity promoted the hydrolysis of the hydrophobic agent. This alkalinity was conducive to the polymerization of OH^−^ with the hydrophobic agent on the surface of the gypsum crystal and the formation of a hydrophobic film, significantly improving the water resistance of the gypsum.

From Figure 6, we can also find that the intensity of OH^-^ decreased obviously compared with that of the pure phosphogypsum group. According to the Chen and Simon’s studies [20,21], organosilicon had a higher degree of hydrolysis and polymerized gradually, generating an amount of the silicon hydroxyl key, which can bond with C-S-H gel and gypsum crystal and delay the cement hydration. Due to hydroxide consumption, the surface of the phosphogypsum crystal was coated with an organosilicon tax layer, which impeded the migration of free water and the attachment of water on the surface. By adding an amount of cement, the alkalinity can be adjusted to accelerate the hydrolysis and condensation process of the organosilicon compounds during the hydration of the phosphogypsum.

#### 3.4.2. Hydrophobic Agent Micromorphology of the Hydration Product

As Figure 7 showed, SEM was carried out on the gypsum blocks for the following samples: the phosphogypsum reference group, the phosphogypsum +0.3 wt.% hydrophobic agent, the phosphogypsum +0.3 wt.% hydrophobic agent +0.4 wt.% cement, the phosphogypsum +0.3 wt.% hydrophobic agent +0.4 wt.% cement, and the phosphogypsum +0.3 wt.% hydrophobic agent +0.6 wt.% cement. SEM analysis was performed to study the effect of different amounts of the silicone hydrophobic agent and cement on the microscopic morphology of phosphogypsum.

The hydration product of phosphogypsum was calcium sulfate dihydrate. As Figure 7a shows, the calcium sulfate dihydrate in the reference group was long, columnar, and randomly distributed in the matrix. They overlap with each other to provide strength for the hardened material. Figure 7b shows that the addition of the hydrophobic agent had no obvious effect on the morphology of the calcium sulfate hydrate, and the hydration products in the matrix also did not change. Meanwhile, the hydrophobic agent formed a lamellar layer that adhered to the surface of crystals to improve their waterproofing. The alkalinity of the gypsum system with a 0.3% hydrophobic agent was adjusted by cement; the addition of cement did not change the main hydration product composition of the matrix, and the calcium sulfate dihydrate crystal retained its original shape. Crystals with poor crystallinity were observed as the addition of cement increase the pH of the system [22]. The aluminate phase in the cement reacted with the phosphogypsum to form needle-like ettringite, thereby leading to a lower strength. The amount of cement increased, the calcium sulfate dihydrate coarsened because the cement promoted the hydrolysis and polymerization reactions of the hydrophobic agent, according to the results of the XPS spectrum. The hydrophobic agent adsorbed onto the surface of gypsum crystals and made the crystals denser. This prevents the water from flowing and migrating into the gypsum, thereby reducing the water absorption.

## 4. Conclusions

Based on the experimental results, some conclusions were obtained as follows:(1)With the increase of the silicone hydrophobic agent dosage from 0.1 wt.% to 0.5 wt.%, the water absorption of the hardened phosphogypsum was decreased from 18.0% to 0.9%. The softening coefficient was improved to 0.99 when the dosage was 0.5 wt.%. Meanwhile, the compressive strength and flexural strength of phosphogypsum decreased first at 0.1 wt.% and then increased as the amount of added hydrophobic agent increased.(2)In the case of 0.3% content of the silicone hydrophobic agent, with the increase in the weight ratios of cement, the softening coefficient of the phosphogypsum was increased from 0.80 to 0.99. The water absorption of the phosphogypsum decreased to 0.8%, with a weight ratio of cement which was 0.6 wt.%. The contact angle of phosphogypsum increased from 47.56° to 147.23° under the coupling effect of hydrophobic agents and cement. However, the compressive and flexural strength of phosphogypsum reduced consistently with the increase of the weight ratios of cement.(3)When the cement was added together with the hydrophobic agent, appropriate alkalinity promoted the hydrolysis of the hydrophobic agent, which was conducive to the polymerization of the hydrophobic organic on the surface of the gypsum crystal. When the 0.3 wt.% of cement was used to adjust alkalinity, gypsum crystal coarsening occurred in the silicone hydrophobic agent–gypsum system. The coarsening of the crystals led to a strength decrease in phosphogypsum. Thus, the coupling of the organosilicon hydrophobic agent and cement exhibited higher mechanical and water-resistance properties than separate organosilicon addition.

Further investigations are needed to understand the underlying coupling effects of other dosages of hydrophobic agent and the weight ratios of cement, instead of being limited to 0.3% silicone dosages. More importantly, the polycondensation of organosilicon, to promote water-resistance, is strongly worth considering to apply to other solid waste materials.

## Figures and Tables

**Figure 1 materials-15-00845-f001:**
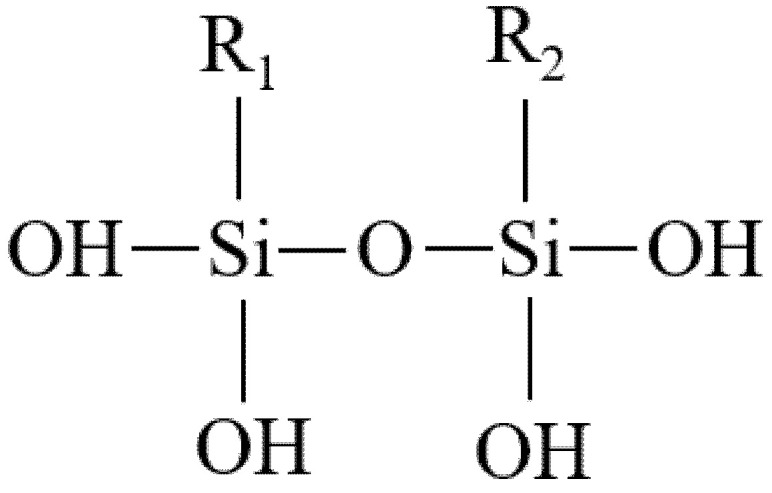
Molecule structure of organosilicon. (R_1_ and R_2_ stand for organic groups.)

**Figure 2 materials-15-00845-f002:**
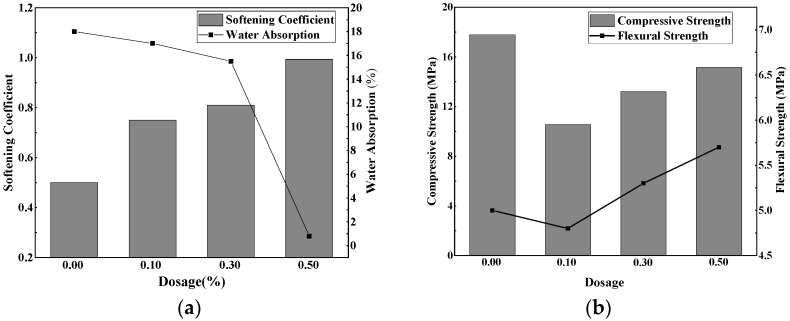
Influence of different hydrophobic agent dosages on the phosphogypsum at 7d. (**a**) Softening coefficient and water absorption rate; (**b**) Compressive strength and flexural strength.

**Figure 3 materials-15-00845-f003:**
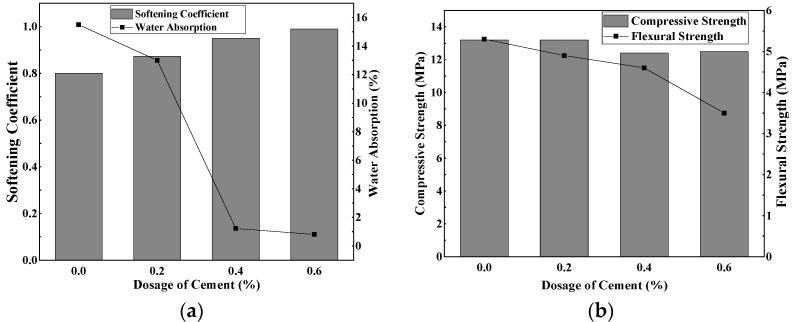
Influence of different cement contents on the phosphogypsum with a hydrophobic agent. (**a**) Softening coefficient and water absorption. (**b**) Compressive strength and flexural strength.

**Figure 4 materials-15-00845-f004:**
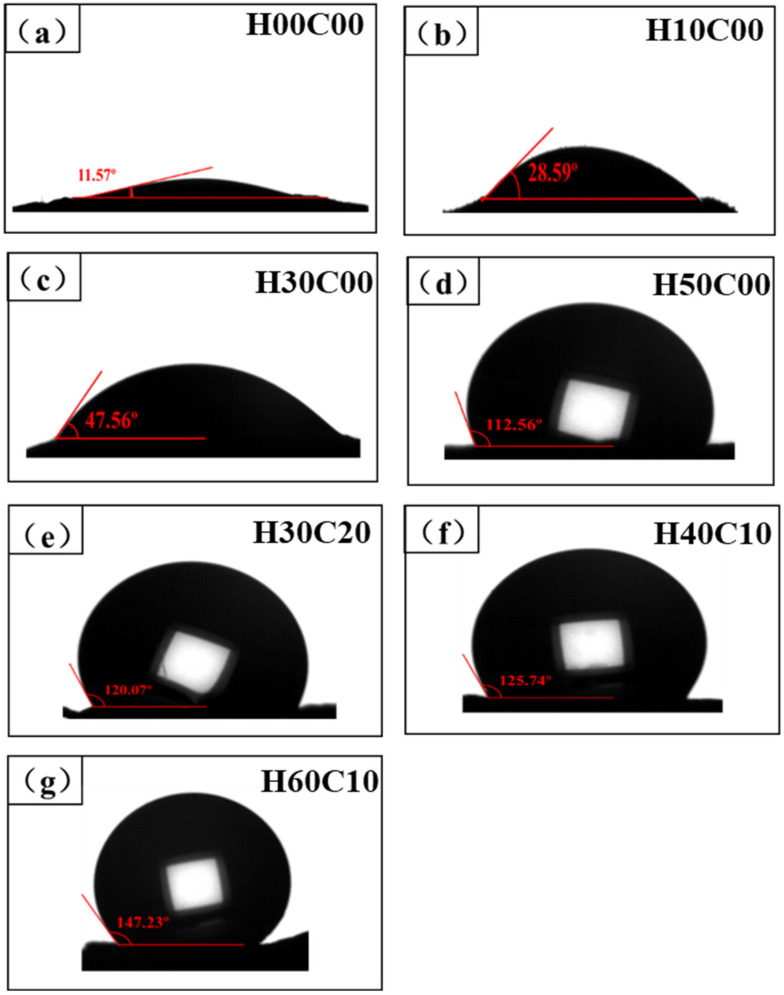
Influence of silicone hydrophobic agent on the phosphogypsum contact angle. (**a**–**g**) Interface contact angles of phosphogypsum blocks with different organic silicon contents.

**Figure 5 materials-15-00845-f005:**
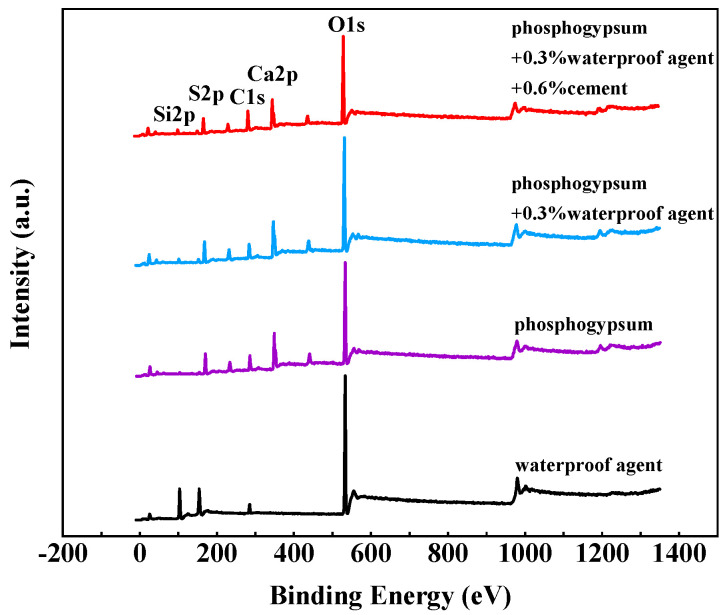
Total XPS spectrum of gypsum sample.

**Figure 6 materials-15-00845-f006:**
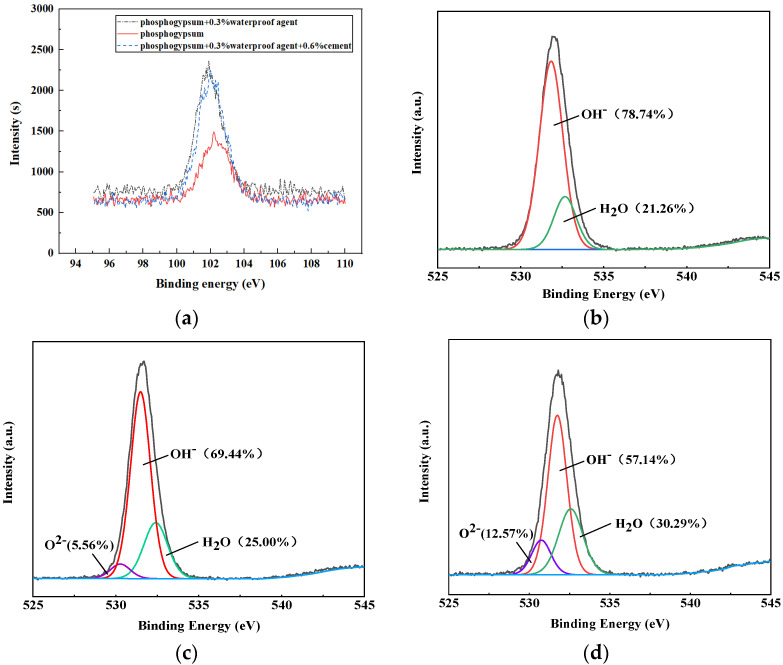
XPS spectra of Si 2p and O 1s in the gypsum sample. (**a**) Change in Si 2p in gypsum samples. (**b**) Phosphogypsum reference group O 1s. (**c**) Phosphogypsum +0.3 wt.% of hydrophobic agent. (**d**) Phosphogypsum +0.3 wt.% hydrophobic agent +0.6 wt.% cement.

**Figure 7 materials-15-00845-f007:**
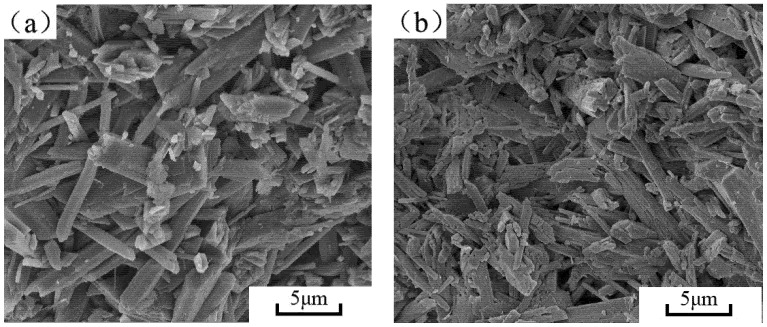
SEM image of plaster sample cross-sections. (**a**) Reference group; (**b**) phosphogypsum +0.3 wt.% hydrophobic agent; (**c**) phosphogypsum +0.3 wt.% hydrophobic agent +0.4 wt.% cement; (**d**) phosphogypsum +0.3 wt.% hydrophobic agent +0.6 wt.% cement.

**Table 1 materials-15-00845-t001:** Chemical composition of phosphogypsum and cement (wt. %).

Oxide	SiO_2_	MgO	Al_2_O_3_	Fe_2_O_3_	CaO	K_2_O	SO_3_	TiO_2_	P_2_O_5_	Loss
Phosphogypsum	4.62	0.18	2.90	1.13	42.67	0.25	43.04	0.21	2.11	2.89
Cement	21.39	2.82	5.15	3.86	61.04	0.62	3.10	0.85	0.10	1.07

**Table 2 materials-15-00845-t002:** Mix design of modified phosphogypsum.

Number	Water-Binder Ratio	Phosphogypsum/%	Cement/%	Hydrophobic Agent/%	Retarder/%	Water Reducing Agent/%
H00C00	0.4	100.0	0.0	0.0	0.1	0.7
H10C00	0.4	100.0	0.0	0.1	0.1	0.7
H30C00	0.4	100.0	0.0	0.3	0.1	0.7
H50C00	0.4	100.0	0.0	0.5	0.1	0.7
H30C20	0.4	99.8	0.2	0.3	0.1	0.7
H30C40	0.4	99.6	0.4	0.3	0.1	0.7
H30C60	0.4	99.4	0.6	0.3	0.1	0.7

**Table 3 materials-15-00845-t003:** Influence of the hydrophobic agent content on the fluidity and setting time of the phosphogypsum slurry.

Number	Hydrophobic Agent/wt.%	Liquidity/mm	Initial Set/min	Final Set/min
H00C00	0	220	45	61
H10C00	0.1	225	47	64
H30C00	0.3	228	49	65
H50C00	0.5	230	50	65

**Table 4 materials-15-00845-t004:** Effect of cement content on fluidity and setting time of phosphogypsum slurry.

Number	Cement Content/wt.%	Liquidity/mm	Initial Set/min	Final Set/min
H30C20	0.2	230	52	67
H30C40	0.4	233	53	69
H30C60	0.6	235	55	70

**Table 5 materials-15-00845-t005:** Contents of each element in gypsum samples.

Samples	Ca 2p/%	C 1s/%/%	O 1s/%/%	Si 2p/%	S 2p/%
Hydrophobicing agent	0.00	9.57	56.52	33.91	0.00
phosphogypsum	10.65	20.89	50.74	3.57	14.15
Phosphogypsum +0.3 wt.% hydrophobic agent	9.99	21.24	51.28	4.85	12.44
Phosphogypsum +0.3 wt.% Hydrophobic agent +0.6 wt.% cement	8.78	29.53	45.39	5.92	10.38

## Data Availability

Data sharing is not applicable to this article.

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
