# Peer review of "The Coupling Effect of Organosilicon Hydrophobic Agent and Cement on the Water Resistance of Phosphogypsum"

_materials, 2022, doi:10.3390/ma15030845_

Round 1

Reviewer 1 Report

The paper investigates an interesting topic such as the coupling effect of cement and organosilicon hydrophobic agents on the water resistance of phosphogypsum. Different dosages of Portland cement were assessed by adjusting the alkalinity of this system and further improve the work efficiency of the organosilicon hydrophobic agents The Methodology is pertinent, the structure of the paper needs to be improved. English is good. There are several points to be developed.
1. The introduction needs to define explicitly the novelties of the paper.
2. The last paragraph of the introduction should describe the structure of the paper.
3. Section 2 describes the methodology that needs also to be expanded and described in details. You may change the title of the section into 'methodology'. 4. Section 3 is overdimensioned and needs to be splitted in two parts: results and discussion.
5.Conclusion needs a final sentence that sum up the three previous points and propose future investigations

Author Response

Reviewer 1:

Q1: The introduction needs to define explicitly the novelties of the paper.

Answer:

Thank you for your advice. This paper aimed to study the coupling effect of cement and organosilicon hydrophobic agents on the water-resistance of phosphogypsum. This is also the innovation of paper. Detail description is present in the revised manuscript. (Line53~Line59).

Q2. The last paragraph of the introduction should describe the structure of the paper.

Answer:

Thank you for your instructive suggestions. We have made detailed revisions in the last paragraph of the introduction (Line60~Line75).

Q3. Section 2 describes the methodology that needs also to be expanded and described in detail. You may change the title of the section into 'methodology'.

Answer:

Thank you for your instructive suggestions. The title of section 2 has been changed into the “Methodology”. Corresponding details have been added in this section. (Line77~Line86, Line 94~Line144).

Q4. Section 3 is overdimensioned and needs to be splitted in two parts: results and discussion.

Answer:

Thank you for your sincere advice. Most manuscripts in the Materials do not separate results and discussion into two parts [1-3]. Meanwhile, the discussion of results is more closely linked to the experimental results in our research. After much deliberation, the authors have decided to present the results and discussion in one part.

Reference

[1] Y.R. Zhang, H. Zhang, X. Zhang. Influence of Calcined Flue Gas Desulfurization Gypsum and Calcium Aluminate on the Strength and AFt Evolution of Fly Ash Blended Concrete under Steam Curing. Materials. 2021,14, 7171.

[2] Y.G. Wang, Na. Zhang, Y.Y. Ren, Y.T. Xu, X.M. Liu. Effect of Electrolytic Manganese Residue in Fly Ash-Based Cementitious Material: Hydration Behavior and Microstructure. Materials. 2021,14,7047.

[3] J.S. Kim, S.J. Kim, K.J. Min, J.C. Choi, H.S. Eun, B.K. Song. Article A Study on Tensile Behavior According to the Design Method for the CFRP/GFRP Grid for Reinforced Concrete. Materials. 2022,15,357.

Q5.Conclusion needs a final sentence that sum up the three previous points and propose future investigations.

Answer:

Thank you for your instructive suggestions. The contents have been added to the modified mansucropt. (Line406~Line413).

Reviewer 2 Report

  1. Manuscript needs proofreading.
  2. One of the most important sides of this article is using hydrophobic materials. Authors have mentioned some general information about those materials in the introduction part, which is considered insufficient. Authors need to give more explanation about hydrophobic materials and their effect, and explain more about specific and common types of such materials, like: silane, siloxane, fluoropolymers, silicon resin. The following articles might be useful: https://doi.org/10.1002/suco.202000214.
  3. In section 2.1, authors only mention that the used material is organosilicon hydrophobic material, which is scientifically not enough. Authors need to specify the type of the material (what general chemical composition it has?).
  4. In section 2.2, authors need to specify the number of tested specimens.
  5. In section 3.1, authors mentioned that: "This was because the hydrophobic agent was adsorbed onto the surface of the gypsum crystals, hindering the hydration of gypsum and leading to a decrease in the compressive strength". Please, use a reference to this information. 
  6. In section 3.3, authors did not mention when did they took the images of contact angles. Was it after 0 seconds or 30 seconds or 60 seconds from starting the test? and why did you take images only for one time period? This test needs to be observed for longer periods starting from 0 sec and extends to 120 sec and sometimes more. refer to the testing procedure of this test: https://doi.org/10.1080/10298436.2019.1567917
  7. In section 3.4, authors explained the SEM analysis of samples. However, they did not explain what is the effect of the changed morphology on the strength and water absorption of samples? They should link the morphological changes with the strength and water absorption.

Author Response

Reviewer 2:

Q1. Authors have mentioned some general information about those materials in the introduction part, which is considered insufficient. Authors need to give more explanation about hydrophobic materials and their effect, and explain more about specific and common types of such materials, like: silane, siloxane, fluoropolymers, silicon resin.

Answer:

Thank you for your instructive advice. The description of materials type and effects are added in the introduction section. (Line40~Line43)

Q2. In section 2.1, authors only mention that the used material is organosilicon hydrophobic material, which is scientifically not enough. Authors need to specify the type of the material (what general chemical composition it has?).

Answer:

Thank you for your sincere advice. The details have been added to the respective pragraph. (Line83~Line86).

Q3. In section 2.2, the authors need to specify the number of tested specimens.

Answer:

Thank you for your sincere advice. The information has been added to the respective pragraph. (Line114~Line120).

 Q4. In section 3.1, authors mentioned that: "This was because the hydrophobic agent was adsorbed onto the surface of the gypsum crystals, hindering the hydration of gypsum and leading to a decrease in the compressive strength". Please, use a reference to this information.

Answer:

Thank you for your sincere advice. The reference has been added to the respective sentence. (Line155).

4. In section 3.3, authors did not mention when did they took the images of contact angles. Was it after 0 seconds or 30 seconds or 60 seconds from starting the test? and why did you take images only for one time period? This test needs to be observed for longer periods starting from 0 sec and extends to 120 sec and sometimes more. refer to the testing procedure of this test: https://doi.org/10.1080/10298436.2019.1567917

Answer:

Thank you for your sincere suggestions. The test of contact angle is based on the standard GB/T 30447-2013 “Measurement method for contact angle of nano-film surface”, and the change of water droplets was observed after 60 seconds from starting the test. The detailed information has been added in section 2.3 of the updated manuscript. The results of the different contact angles in Fig. 4 already allow a valid analysis of the differences in the water resistance of the specimens with different organic silicone and cement contents. Meanwhile, some researchers have also selected the contact angle results at a fixed time when conducting contact angle test [1-3].

[1] Li J , Cao J , Ren Q , et al. Effect of nano-silica and silicone oil paraffin emulsion composite waterproofing agent on the water resistance of flue gas desulfurization gypsum[J]. Construction and Building Materials, 2021, 287(8):123055.

[2] Xy A , Hl B , Hua D C , et al. Influence of water repellent on the property of solid waste based sulfoaluminate cement paste and its application in lightweight porous concrete[J]. Construction and Building Materials, 2021,282:122731.

[3] Bo Wu , JS Qiu . Enhancing the hydrophobic PP fiber/cement matrix interface by coating

nano-AlOOH to the fiber surface in a facile method [J].Cement and Concrete Composites, 2022,125:104297.

5. In section 3.4, authors explained the SEM analysis of samples. However, they did not explain what is the effect of the changed morphology on the strength and water absorption of samples? They should link the morphological changes with the strength and water absorption.

Answer:

Thank you for your instructive suggestions. In section 3, the link between SEM analysis and strength and water absorption has been established. The detailed modification can be found at line 660- line 662 and line 665- line 672 in the updated manuscript.

Reviewer 3 Report

This article investigates experimentally the cumulative effect of cement and organosilicon hydrophobic on the resistance of phosphogypsum to water absorption among other tested parameters. The extent of the experimental work by the authors is praised. The quality of the experimental work and presentation of the results are acceptable. This reviewer believes the manuscript merits being published as a journal article provided the following comments are addressed.

Line 10: You might want to consider using "weight ratio" rather than "dosage" here and throughout the manuscript.

Line 22, Line 54, Line 82, Line 83: Please remove the hyperlink.

Line 31 and Line 35: Please rephrase this sentence since it is not clear what the main point of the sentence is.

Line 61 - 64: This large sentence can be divided into two sentences for the sake of clarity.

Line 94, Table 2: The second decimal can be removed and the table font must be in black.

Line 202: Do you mean the 7th day or day 7?

Line 243 Figure 4: Is there a way you can quantify Intensity on the Y-axis of this figure?

Line 266 Figure 5: Same as above. Please write the caption for Figure 5a in black.

Line 343: This reviewer suggests that all the observations and experiments get separated from the discussions. The authors might want to add a separate section under the discussion heading.

Line 360: Please remove the hyperlink.

Author Response

Reviewer 3

Q1 Line 10: You might want to consider using "weight ratio" rather than "dosage" here and throughout the manuscript.

Answer:

Thank you for your sincere suggestions. The use of “dosage” has been changed to “weight ratio.” Moreover, I have highlighted all corrections with the “editing mode” in the updated manuscript.

Q2 Line 22, Line 54, Line 82, Line 83: Please remove the hyperlink.

Answer:

Thank you for your sincere suggestions. All the hyperlink has been removed in the updated manuscript.

Q3 Line 31 and Line 35: Please rephrase this sentence since it is not clear what the main point of the sentence is.

Answer:

Thank you for your sincere advice. I have redrafted the line 31 to 35 with the “editing mode” in the updated manuscript. The details of modification are listed below:

“Although some attempts have been made to use phosphogypsum as a wallboard and building blocks in the field of construction[2-4], the poor water-resistance of phogypsum severely restricts its widespread application[5].”

Q4 Line 61 - 64: This large sentence can be divided into two sentences for the sake of clarity.

Answer:

Thank you for your sincere advice. The sentence in line 61 – 64 has been divided into two sentences. The details of modification are listed below:

“It would be substantiated that the coupling of organosilicon hydrophobic agent and cement content is a more effective way to modify the water-resistance phosphogypsum. The enhancement of phosphogypsum's water-resistance can help accelerate the process of phosphogypsum solid waste utilization.”

Q5 Line 94, Table 2: The second decimal can be removed and the table font must be in black.

Answer:

Thank you for your instructive suggestions. In Table 2, The second decimal has been removed, and the table font has been modified to black.

Q6 Line 202: Do you mean the 7th day or day 7?

Answer:

Thank you for your question. “at 7d” means all tests were conducted after gypsum curing after 7 days. After further consideration, “at 7d” has been removed from the updated manuscript as the relevant information for tests has been given in section 2.3.

Q7 Line 243 Figure 4: Is there a way you can quantify Intensity on the Y-axis of this figure?

Answer:

Thank you for your sincere suggestion. For Fig. 4, the ordinate has no dimension because it clearly qualitatively compares intensity differences in element composition.

Q8 Line 266 Figure 5: Same as above. Please write the caption for Figure 5a in black.

Answer:

Thank you for your sincere advice. The caption for Figure 5a has been modified in black. The dimensional diagram is shown in the attached Fig. 5a.

Q9 Line 343: This reviewer suggests that all the observations and experiments get separated from the discussions. The authors might want to add a separate section under the discussion heading.

Answer:

Thank you for your instructive suggestions. Most manuscripts in the Materials do not separate results and discussion into two parts[1-3]. Meanwhile, the discussion of results is more closely linked to the experimental results in our research. After much deliberation, the authors have decided to present the results and discussion in one part.

Reference

[1] Y.R. Zhang, H. Zhang, X. Zhang. Influence of Calcined Flue Gas Desulfurization Gypsum and Calcium Aluminate on the Strength and AFt Evolution of Fly Ash Blended Concrete under Steam Curing. Materials. 2021,14, 7171.

[2] Y.G. Wang, Na. Zhang, Y.Y. Ren, Y.T. Xu, X.M. Liu. Effect of Electrolytic Manganese Residue in Fly Ash-Based Cementitious Material: Hydration Behavior and Microstructure. Materials. 2021,14,7047.

[3] J.S. Kim, S.J. Kim, K.J. Min, J.C. Choi, H.S. Eun, B.K. Song. Article A Study on Tensile Behavior According to the Design Method for the CFRP/GFRP Grid for Reinforced Concrete. Materials. 2022,15,357.

Q10 Line 360: Please remove the hyperlink.

Answer:

Thank you for your sincere suggestions. The hyperlink has been removed in the updated manuscript.

Round 2

Reviewer 1 Report

Ready for publication

Reviewer 2 Report

Authors have addressed most of the comments. However, paper is now satisfactory and can be accepted